# Anatomical Targeting of Anticancer Drugs to Solid Tumors Using Specific Administration Routes: Review

**DOI:** 10.3390/pharmaceutics15061664

**Published:** 2023-06-06

**Authors:** Akira Saito, Joji Kitayama, Ryozo Nagai, Kenichi Aizawa

**Affiliations:** 1Department of Surgery, Jichi Medical University, 3311-1 Yakushiji, Shimotsuke, Tochigi 329-0431, Japan; saito.akira@jichi.ac.jp (A.S.); kitayama@jichi.ac.jp (J.K.); 2Division of Translational Research, Clinical Research Center, Jichi Medical University Hospital, Tochigi, Tochigi 329-0498, Japan; 3Department of Medicine, School of Medicine, Jichi Medical University, 3311-1 Yakushiji, Shimotsuke, Tochigi 329-0498, Japan; rnagai@jichi.ac.jp; 4Division of Clinical Pharmacology, Department of Pharmacology, Jichi Medical University, 3311-1 Yakushiji, Shimotsuke, Tochigi 329-0498, Japan

**Keywords:** anticacncer dryg, intra-arterial route, intrathecal route, intrapleural route, intraperitoneal route, intratumor route, thoracic duct infusion

## Abstract

Despite remarkable recent progress in developing anti-cancer agents, outcomes of patients with solid tumors remain unsatisfactory. In general, anti-cancer drugs are systemically administered through peripheral veins and delivered throughout the body. The major problem with systemic chemotherapy is insufficient uptake of intravenous (IV) drugs by targeted tumor tissue. Although dose escalation and treatment intensification have been attempted in order to increase regional concentrations of anti-tumor drugs, these approaches have produced only marginal benefits in terms of patient outcomes, while often damaging healthy organs. To overcome this problem, local administration of anti-cancer agents can yield markedly higher drug concentrations in tumor tissue with less systemic toxicity. This strategy is most commonly used for liver and brain tumors, as well as pleural or peritoneal malignancies. Although the concept is theoretically reasonable, survival benefits are still limited. This review summarizes clinical results and problems and discusses future directions of regional cancer therapy with local administration of chemotherapeutants.

## 1. Introduction

In recent years, prognoses of patients with metastatic cancer have improved due to advances in chemotherapeutic drugs and emergence of new molecular targets [1,2]. However, outcomes of cancer patients, especially those with solid tumors, still remain unsatisfactory. Anticancer agents are usually administered though peripheral veins, since most anti-cancer agents are easily absorbed in blood and rapidly carried throughout the body. Despite being the most common route of drug administration, systemic chemotherapy, especially with dose-escalation strategies, often has limited anti-cancer effects because of hematologic and non-hematologic toxicities [3,4]. Indeed, distribution of systemically administered drugs depends largely on the blood supply of each organ. As a result, healthy organs may be subjected to high drug doses while insufficient doses are delivered to target lesions, producing high systemic toxicity with minimal anti-tumor efficacy. In such cases, it is necessary to develop drug transporters, e.g., nanoparticles [5,6], that can reach the target more efficiently or to develop more efficient drug delivery methods in order to enhance therapeutic effectiveness.

An alternative approach is regional administration of anti-cancer drugs. Because the route of administration directly affects drug bioavailability, which determines both onset and duration of pharmacological effects in targeted organs, various types of local-regional chemotherapy have been attempted [7,8,9,10]. Among them, intra-arterial (IA), intrathecal (ITh), intrapleural (I-Pl), intraperitoneal (I-Pr) and intratumoral (IT) routes have been reported as useful treatment strategies for specific tumors (Figure 1). This review provides a comprehensive overview of clinical results and challenges of current anti-cancer therapies using various routes of administration, and discusses the future of local cancer therapy.

## 2. Intra-Arterial Route (IA)

This regional approach is suitable for cancers in organs with abundant arterial blood supply. Currently, IA administration of anticancer drugs is performed for hepatocellular carcinomas (HCC), metastatic liver tumors, brain tumors, and head and neck cancers.

### 2.1. HCC

A wide range of therapeutic options is available for HCC. Outcomes for patients undergoing liver resections are generally unfavorable, as a result of the high postoperative recurrence rate [11]. Chemotherapy is one of the most important treatment modalities for advanced HCC. However, the efficacy of chemotherapy is still unsatisfactory, resulting in poor prognoses [12,13,14,15,16,17]. As progression of HCC is largely dependent on blood flow from the hepatic artery, transarterial chemoembolization (TACE) and hepatic arterial infusion chemotherapy (HAIC) are often used in clinical practice, and these greatly improve therapeutic effects.

TACE is one of the standard treatment modalities. In the international guideline-approved Barcelona Clinic Liver Cancer (BCLC) staging system, TACE is considered the standard of care for intermediate-stage HCC, including unresectable, multinodular HCC without extrahepatic metastases [18,19]. TACE consists of an intra-arterial injection of anti-cancer drugs emulsified in Lipiodol, an oily and radiopaque agent, followed by an additional intra-arterial injection of an embolic agent, such as a gelatin sponge. By this method, Lipiodol allows the anti-cancer drug to reach the tumor directly, causing embolization of tumor microcirculation, enhancing antitumor efficacy. Moreover, Lipiodol that remains in the tumor can be detected by image inspections even after treatment, making it possible to determine therapeutic efficacy.

Drug-eluting beads (DEBs), non-resorbable, embolic microspheres, can release drugs. By using DEB, sustained tumor effects of cytotoxic agents with tumor embolic effects were maintained, and their efficacy has recently been evaluated [20,21,22,23]. Irie et al. reported development of a technique called balloon-occluded TACE (B-TACE) [24]. B-TACE is defined as infusion of a chemotherapeutic emulsion containing Lipiodol, followed by infusion of a gelatin particle sponge with occlusion of the feeding artery using a microballoon catheter [25]. Occlusion of the feeding artery containing Lipiodol at the target nodule, results in higher accumulation of the chemotherapeutic emulsion. In recent years, several reports have shown that the therapeutic effect of B-TACE is superior to that of conventional TACE [26,27,28]. In the future, it may offer a safe and effective alternative to the current standard catheter TACE [29].

HAIC is employed to treat patients who are deemed unsuitable candidates for surgical resection, local ablative therapy, or TACE, that is, patients who have extrahepatic metastasis, show evidence of vascular invasion, or are refractory to TACE [12,13,14,15,16,17]. HAIC permits chemotherapeutic agents to be delivered directly into feeding arteries of liver tumors while maintaining high local drug concentrations using an implantable port system. Therefore, compared with systemic chemotherapy, it is possible to enhance anti-tumor effects and to minimize systemic toxicity [30]. Several reports have suggested that HAIC improves overall survival (OS) and progression-free survival (PFS) in patients with advanced HCC, suggesting that HAIC is more effective than conventional systemic chemotherapy [31,32,33,34,35]. In the future, the efficacy of HAIC combined with systemic chemotherapy or molecularly targeted drugs will be investigated [36,37], and if a survival benefit is demonstrated, HAIC will be recognized as a standard treatment for patients with advanced HCC [38].

### 2.2. Metastatic Liver Tumors

Colorectal cancer (CRC) is one of the leading cancers globally in terms of both incidence and mortality [39]. Liver metastatic disease invariably results from metastasis in colorectal cancer patients due to the fact that venous drainage from the colon and rectum allows metastases to migrate to the liver via the portal vein. Because of its high prevalence, liver-directed therapies have been developed, and hepatic arterial infusion (HAI) is now used for metastatic liver tumors as well.

In patients with unresectable metastases to the liver, HAI combined with systemic chemotherapy, increases the likelihood of resection to provide patients with a chance for cure, compared with systemic therapy alone. It shows increased reaction and conversion [40]. In patients who receive HAI in an adjuvant setting after liver surgery, HAI therapy given with systemic chemotherapy can increase disease-free survival [41].

### 2.3. Brain Tumors

First-pass metabolism of intravenously (IV)-delivered drugs and the blood-brain barrier (BBB) make it difficult to achieve therapeutically effective concentrations of anti-neoplastic agents against brain tumors. These are the biggest obstacles to treating brain tumors [42]. IA chemotherapy is a form of regional delivery to brain tumors, designed to enhance intra-tumoral concentrations of drugs, in comparison with the intravenous route. IA delivery directly to the cerebral vasculature, obviates first-pass metabolism and associated systemic adverse effects of IV chemotherapy. Recently, clinical studies have shown therapeutic efficacy of IA chemotherapy for low- and high-grade gliomas and cerebral lymphomas. Additionally, carboplatin and methotrexate can reduce toxicity [43,44], but no Phase III trials have been conducted with carboplatin or methotrexate. Further clinical studies are needed to establish IA chemotherapy in treatment of primary brain tumors.

### 2.4. Head and Neck Cancers

Advanced head and neck cancers typically involve multiple organs, such as the larynx, pharynx, tongue, and maxilla, and treatment strategies, including partial or complete resection have been considered. However, while these are curative treatments for cancer, they still pose major problems as they inevitably lead to substantial functional impairment and disfigurement [45]. The desired treatment for head and neck tumors is eradication of the tumor while preserving function and appearance. Therefore, in order to improve the survival rate of patients with advanced head and neck cancer without sacrificing function, multidisciplinary treatment including, not only surgery, but also radiotherapy and systemic chemotherapy is needed. Blood is supplied to head and neck tumors mainly from branches of the external carotid artery, and IA chemotherapy, which is specialized for local chemotherapy, is a suitable treatment for localized malignant neoplasms of head and neck cancer. Recently, significant advances in vascular radiology techniques and development of new devices, such as fluoroscopy units and angiographic catheters, have enabled repetitive hyperselective IA chemotherapy. IA infusion of high-dose cisplatin with systemic neutralization and intravenous sodium thiosulfate for advanced head and neck cancer has become a therapeutic modality with low systemic toxicity and high tumor response [46]. In addition, Heianna et al. suggested that selective intra-arterial chemoradiotherapy with docetaxel and nedaplatin may achieve both good local control and survival in bulky, node-fixed head and neck cancer of unknown primary origin (HNCUP) [47] (Table 1).

## 3. Intrathecal Route (ITh)

The incidence of metastatic brain tumors from various cancers ranges from 9% to 30% [48,49,50,51,52]. Melanoma, breast cancer, and lung cancer are the main carcinomas that cause brain metastasis [48], and metastatic brain tumors generally have poor prognosis [51]. Conventional systemic anti-cancer treatments, including chemotherapy and targeted therapies, are largely ineffective against metastatic brain tumors. This is because there are three barrier systems (arachnoid, blood-cerebrospinal fluid (BCSFB), and blood-brain barrier (BBB)) that physically and functionally separate extracellular fluids from the central nervous system (CNS). Therefore, anti-cancer drugs fail to reach metastatic brain tumors at effective concentrations [48,50,53,54,55]. For most antineoplastic agents, total cerebrospinal fluid (CSF) exposure following administration of a systemic dose is less than 10% of systemic exposure. Furthermore, since there is a BBB after that, most anticancer drugs do not reach the brain [56]. Therefore, ITh administration was developed with the hope of effectively delivering anticancer drugs to brain tumors. ITh delivery injects substances directly into CSF-containing spaces in the CNS. Anti-cancer drugs can be delivered to the subarachnoid space by three ITh methods (lumbar injection, cerebellar medullary cistern injection, or injection into the ventricular system) [57]. ITh administration allows transport of anti-cancer drugs between the BBB and BCSFB, enhancing drug concentrations in the much smaller volume of CSF (compared to plasma). This allows drug doses to be reduced while maintaining drug concentrations in the CNS and minimizing systemic toxicity [57,58]. ITh drug delivery involves injection into the lateral ventricles via a subcutaneous reservoir called an Ommaya reservoir and a ventricular catheter [59]. The Ommaya reservoir is a subcutaneous device, with a catheter inserted into one of the lateral ventricles of the brain, providing direct access to ventricular CSF [60]. “CNS prophylaxis” with repeated cycles of ITh methotrexate (MTX) has replaced CNS prophylactic irradiation in children with low-risk acute lymphoblastic leukemia, drastically reducing the incidence of CNS relapse from 50% to 23% [61].

Donovan et al. suggest that repeat administration of CAR-T-cells, perhaps through an Ommaya reservoir, could increase therapeutic efficacy, compared to either IV administration, or single dose intraventricular administration via the lateral ventricle (LV). Delivery of CART-cell therapy directly into the CSF likely increases the exposure of CAR-T-cells to cancer cells and may decrease systemic toxicity. From the above, Donovan et al. suggested that locoregional delivery of CAR-T-cells directly into the CSF may reduce risk of systemic toxicities associated with CAR-T-cells, in comparison to the more commonly used intravenous approach [62].

In addition, ITh was investigated not only for brain tumors, but also for leptomeningeal metastases (LM). Rhun et al. reported a clinically meaningful gain in LM-related PFS when breast cancer patients with newly diagnosed LM received intrathecal liposomal cytarabine chemotherapy together with systemic treatment, compared with systemic treatment alone [63].

Finally, we present another administration route for brain tumors, convection-enhanced delivery (CED), which is slightly different from the intrathecal route. CED is a new drug administration method that uses a pressure-driven catheter to locally inject drugs into intercellular spaces of the brain under continuous positive pressure to achieve high concentration and wide drug distribution [64]. Specifically, this catheter is stereotactically placed into the tumor tissue via a burr hole under magnetic resonance imaging (MRI) guidance. The catheter is then connected to an extracranial infusion pump to distribute the infusate to the tumor by convective transport. Thus, therapeutic agents can penetrate tissue by several centimeters from the catheter tip in a pseudo-spherical distribution, in contrast to only a few millimeters with diffusion-dependent injection modalities [65]. Considering that the vast majority of brain tumor recurrences occur within 2 cm of the tumor border, the area of drug penetration after dosing completely permeates this tissue [66,67,68,69]. Currently, many clinical trials have been conducted with this administration method, and favorable results have been reported (Table 2).

## 4. Intrapleural Route (I-Pl)

Clinical situations in which drugs are administered into the pleural cavity by the I-Pl route, include intrapleural fibrinolytic therapy for empyema [71], pleurodesis for recurrent pneumothorax and recurrent pleural effusion [72], and I-Pl administration of anticancer drugs for malignant pleural effusion (MPE).

MPE is the most serious complication of non-small cell lung cancer (NSCLC). MPE occurs in approximately 15% of patients with NSCLC and 50% of these patients eventually develop MPE [73,74]. MPE results in symptoms such as chest discomfort, shortness of breath, palpitations, pain, and an inability to lie down, which significantly reduces patient quality of life [75,76,77]. MPEs have poor prognoses, with a median survival time of 3 to 12 months [78]. Traditional treatments for MPE include systemic chemotherapy, targeted therapy, immunotherapy, and locoregional therapy [79]. Among them, locoregional therapy for MPE is the local perfusion of talc, chemotherapeutic agents, biologic agents, and antiangiogenic agents into the pleural space to achieve adhesions in the pleural cavity [80,81,82]. Intracavity infusion of drugs after removal of pleural effusion is a standard treatment for symptomatic MPE. Drugs administered intrapleurally have been used as cytotoxics, biological response modifiers, and sclerosing agents. However, efficacies and toxicities are unsatisfactory [83].

In recent years, however, I-Pl administration of various anticancer drugs has been investigated for MPE of lung cancer, with hopeful results. Song et al. compared efficacy of I-Pl infusion of Bevacizumab (BEV) and pemetrexed with BEV and cisplatin in MPE caused by NSCLC. The objective response rate (ORR) and disease control rate (DCR) of patients treated with I-Pl infusion of BEV combined with pemetrexed was superior to that of those treated with BEV and cisplatin. The BEV and pemetrexed group also showed statistical improvement in PFS compared with the group treated with BEV and cisplatin [84].

Nie et al. compared efficiency and toxicities of I-Pl and IV infusion of bevacizumab for MPE mediated from non-squamous NSCLC in order to reveal the relationship between serum VEGF levels and outcomes of pleural effusion in NSCLC. The result was that I-Pl infusion of bevacizumab had a higher objective response rate (ORR), longer duration of response (DOR) and less toxicity than IV infusion [85]. Recently, extracellular vesicles labeled as tumor microparticles (TMPs) released by tumor cells are used as natural carriers to deliver chemotherapeutic drugs or oncolytic viruses to tumor cells [86,87]. In particular, I-Pl injection of TMP packaging methotrexate (TMPs-MTX) has proven safe and effective in maintaining anti-tumor effects and in reversing drug resistance [86,88,89,90] (Table 3).

## 5. Intraperitoneal Route (I-Pr)

Metastasis to the peritoneum is a severe complication of abdominal cancers that causes debilitating symptoms and clinical deterioration with poor prognosis. The peritoneum covers the abdominopelvic organs and the physiologic peritoneum-plasma barrier limits uptake of effective concentrations of chemotherapeutics after systemic administration. After intraperitoneal administration, however, the peritoneum-plasma barrier also hinders drug loss to the systemic circulation, facilitating prolonged exposure and higher drug concentrations at the peritoneal surface than in plasma. Therefore, if anti-cancer agents used for IP chemotherapy can be prevented from exiting the peritoneal cavity rapidly, they can achieve greater tumor penetration [91]. Thus, the peritoneum-plasma barrier can be used to enhance locoregional therapeutic efficacy with limited systemic toxicity. The theoretical rationale for I-Pr chemotherapy was first described in 1978 by Dedrick et al., who showed that IP administration results in higher drug concentrations and longer half-lives in the peritoneal cavity, compared with systemic administration [92]. In the past, treatment of peritoneal metastases consisted of systemic chemotherapy or palliative surgery, which were not effective treatments [93].

### 5.1. Heated Intraperitoneal Chemotherapy (HIPEC)

Historically, I-Pr chemotherapy has been performed under hyperthermic conditions (heated intraperitoneal chemotherapy; HIPEC in a single intraoperative procedure that delivers anti-cancer drugs in a heated solution directly to the abdominal cavity after cytoreductive surgery (CRS) [94]. Moderate hyperthermia (41–43 °C) sensitizes tumor cells to DNA-damaging agents, such as platinum compounds and alkylating agents [95,96,97]. Heat also increases tumor cell membrane permeability leading to higher intracellular drug concentrations, and increased penetration of chemotherapeutants at the peritoneal surface [98,99,100]. The procedure to deliver anticancer drugs in a heated solution directly into the peritoneal cavity was first employed by Spratt et al. for treatment of peritoneal pseudomyxoma [101]. Later, Sugarbaker et al. successfully introduced HIPEC in combination with cytoreduction surgery (CRS), because peritoneal metastases (PM) were considered to be lesions confined to the peritoneal cavity [102]. CRS performed in combination with HIPEC is preferentially performed via open rather than laparoscopic surgery. Also, complete cytoreductive surgery requires comprehensive surgical exploration and periodic omentectomy. Peritoneal and organ resection are performed depending on the extent and location of the lesion [103]. Currently, HIPEC combined with CRS is performed mainly for pseudomyxoma, mesothelioma, ovarian, and colorectal cancers, resulting in improved outcomes of patients with peritoneal PM from colorectal [104,105] or ovarian [106,107] cancer, or mesothelioma [108]. Although evidence for efficacy of HIPEC is relatively limited because of its infrequent use in western countries, effective chemotherapeutic regimens and therapeutic effects are being investigated for gastric [109,110] and pancreatic [111] cancer. However, it is likely that HIPEC will offer fewer benefits for patients with PM of gastric or pancreas cancer because of the high-grade metastatic cancer cells. In fact, a recent review suggests that aggressive treatment for gastric cancer should be applied only in cases with a low Peritoneal Carcinoma Index (PCI < 6) [112]. Representative clinical results on HIPEC are summarized in Table 4.

On the other hand, HIPEC with CRS is often associated with serious complications that require intensive management. Gagniere et al. suggested that HIPEC for elderly patients, especially those over 70 years of age, may be associated with more grade 3 or higher complications and deaths; thus, such cases require special attention [113]. The extent to which effectiveness and adverse events of HIPEC are affected by patient selection, choice of intraperitoneal chemotherapeutic drugs, doses, and durations, temperature, and HIPEC regimens, remain largely unknown. Therefore, at present, CRS + HIPEC is performed only on selected patients in specialized facilities, and there are few data from clinical trials comparing it to other treatment methods [114]. It is necessary to perform large-scale, randomized, control trials to optimize and determine the clinical usefulness of HIPEC.

Predictive markers for potential benefit and harm associated with CRS + HIPEC are clinically important for appropriate patient selection. Concomitant lymph node metastasis, liver metastasis, signet ring cell tumor biology, and poor tumor differentiation are poor prognostic factors [115]. Translational research to identify novel molecular and biological markers is a future challenge in this field.

Efficacy of HIPEC in combination with perioperative systemic chemotherapy is being studied in colorectal cancer. The CAIRO6 [116] study is a randomized trial to determine the role of perioperative systemic therapy in addition to CRS + HIPEC for patients with colorectal peritoneal metastases. However, the effect of HIPEC with perioperative systemic chemotherapy remains undetermined. Further investigation is necessary to assess the exact role of CRS + HIPEC combined with perioperative chemotherapy, as it may offer additional clinical benefits.

### 5.2. Pressurized Intraperitoneal Aerosol Chemotherapy (PIPAC)

PIPAC, a method to deliver anti-cancer drugs in aerosolized form created with a nebulizer system, has been proposed as an alternative to HIPEC to improve drug distribution and tissue uptake, as well as enhanced tolerance by patients [117,118]. Using aerosols allows uniform redistribution of substances within an enclosed space. Creating an artificial pressure gradient can offset the tumor interstitial fluid pressure, which is an obstacle to cancer therapy [119,120]. In addition, increasing intraperitoneal pressure particularly enhances uptake of drugs into tumors, resulting in a higher local disposition [117,121,122,123]. Another property leading to superior local disposition is the high drug concentration in the aerosol. Although administered at only 1/10 of the total dose, anticancer drug concentration in the aerosol can be three times higher than that in intraperitoneal fluids typically used in HIPEC, without compromising tolerability [122,124]. Moreover, preclinical studies showed that PIPAC results in good distribution and penetration into tumor nodules in the abdominal cavity [125,126]. Based on these results, PIPAC has been broadly adopted during the past decade, mainly in Europe. Recent reviews suggest that PIPAC is safe and feasible, and offers hope for patients with various types of peritoneal malignancies, although prospective controlled trials are necessary in the future [127,128].

### 5.3. I-Pr Repeated Administration of Taxanes

Another disadvantage of HIPEC is that single-dose administration results in insufficient anticancer agent exposure to peritoneal metastases. Therefore, repeated IP injections are required to obtain a sufficient therapeutic effect on PM. Recently, repeated IP infusion of anti-cancer drugs has become possible using implantable port systems [129,130]. Taxanes such as paclitaxel (PTX) or docetaxel (DOC) are broad-spectrum anticancer drugs that are clinically effective against various types of cancer. Taxanes are theoretically ideal drugs for I-Pr chemotherapy because they stay in the peritoneal space for a long time due to their hydrophobic properties [131], which enables direct penetration into peritoneal disseminated tumors [91,132,133]. However, the depth of infiltration after one-time IP administration of a taxane is limited [134]. In a previous study, we showed that the distance of PTX infiltration was only 100–200 μm beneath the surface of the tumor [133]. Therefore, it is necessary to repeat IP administration to improve antitumor effects to the PM. Fortunately, even if PTX is repeatedly administered intraperitoneally, it rarely causes adhesion of organs in the peritoneal cavity because of its antiproliferative effect, and distribution of i.p. PTX across the intra-abdominal space is not hampered by drug-induced peritonitis [135]. An implantable intraperitoneal access port system is useful for repeated administration of PTX. Once the port is implanted subcutaneously, anti-cancer drugs can be minimally invasive and repeatedly injected into the abdominal cavity without additional invasive surgery. Repeated IP chemotherapy using the port is safe and feasible with proper management and resolution of port complications [136]. In addition, ascites or lavages can be repeatedly collected from the port during each chemotherapeutic cycle, which can provide useful information to assess therapeutic effects.

Recent phase II studies have suggested that I-Pr administration of PTX at a normal temperature, i.e., without heating, is effective for PM from gastric [137,138,139] or pancreatic [140,141] cancer. A randomized, multicenter, phase Ⅲ trial (PHOENIX-GC Trial) [142] was performed to compare I-Pr and IV PTX + S-1 (IP) with the Japanese standard regimen of S-1 + cisplatin (SP) in patients with GC with peritoneal metastasis. Unfortunately, this trial failed to show statistical superiority of IP-PTX + systemic chemotherapy, possibly due to a randomization bias and protocol violations in many patients. However, subsequent exploratory sensitivity analyses (follow-up analysis of 3-year overall survival rate and comparison of treatment responses based on change in ascites among PPS, excluding patients with post-protocol treatment violations) strongly suggest clinical benefits of the IP regimen (Table 5).

Malignant ascites are often present in patients with peritoneal dissemination, seriously affecting the therapeutic efficacy of I-Pr chemotherapy. Reinfusion of autologous ascitic fluids, which contain large amounts of protein and nutrients after filtration and concentration using a special membrane system, is called cell-free and concentrated ascites reinfusion therapy (CART). This has been used with significant clinical benefits for cachexic patients with malignant ascites [143,144], and is especially effective for palliation for patients with symptomatic ascites. In addition, CART often improves performance status of patients who can receive repeated I-Pr administration of PTX, which results in improved survival [145]. CART is now recommended as a treatment option for patients with malignant ascites in Japan [146].
pharmaceutics-15-01664-t005_Table 5Table 5Studies that employed intraperitoneal drug administration (normothemic I-Pr).IndicationsType of CancerAnti-Cancer DrugStudy PhasePeritoneal metastasis (PM)Gastric cancerPaclitaxe [131,133,136,142]Ⅲ  UMIN000005930Gastric cancerDocetaxel [132]-Gastric cancerCatumaxomab [147,148]Ⅱ/Ⅲ  NCT0083664Pancreatic cancerPaclitaxel [134,135,140,141]Ⅰ/Ⅱ  UMIN000018878Ovarian cancerCatumaxomab [147,148]Ⅱ/Ⅲ  NCT0083664Ovarian cancerBevacizumab [149,150]Ⅱ  ANZGOG 11–01Colorectal cancerBevacizumab [149]-Breast cancerBevacizumab [149]-Uterine cancerBevacizumab [149]-


### 5.4. Other Novel Drugs

Recently, novel molecular targeting drugs have been used clinically via IPr routes without heat (Table 5). Catumaxomab is a trifunctional monoclonal antibody with two antigen-binding sites, EpCAM and CD3, and a functional Fc domain that activates a complex antitumor immune reaction through various effector functions, such as antibody-dependent cellular cytotoxicity, phagocytosis, and T cell-mediated cytotoxicity [147]. Heiss et al. performed a randomized phase II/III trial, and reported that IP injection of catumaxomab improved puncture-free survival and exhibited better survival in patients with malignant ascites caused by various malignancies, including gastric cancer [148]. According to their results, catumaxomab, has been licensed for clinical use in the European Union since 2009 for malignant effusion, and promising results have been reported for patients with gastric cancer with PM from gastric cancer [151,152]. Another drug, bevacizumab, a humanized variant of an anti-VEGF antibody, could be useful against malignant ascites [149], since vascular endothelial growth factor A (VEGF-A) is a key mediator of angiogenesis. Sjoquist et al. have shown that IP infusion of bevacizumab is effective for patients with chemotherapy-resistant ovarian cancer with symptomatic ascites [150]. In addition, development of an engineered, exosome-based, peritoneal-localized hydrogel was recently reported to domesticate peritoneal macrophages. Exosomes were fabricated from genetically engineered M1-type macrophages with overexpressed Siglec-10 (SM1Aexo), which were further chemically decorated with sodium alginate oxide (OSA) to form a gelator (O-SM1Aexo). In addition to this, a hydrogel loaded with an efferocytosis inhibitor (MRX-2843) is co-administered intraperitoneally. Administered SM1Axo polarizes M2-type macrophages in the peritoneum to M1-type macrophages, and overexpressed Siglec-10 competitively blocks CD24 on macrophages and amplifies TAM phagocytosis. Furthermore, MRX-2843 enabled enhanced accumulation of dying tumor cells, ensuring adequate release of tumor-derived cGAMP and DAMPs to induce a strong STING-mediated secretion of type I interferon in TAMs and to improve immunogenicity. These factors exert an antitumor effect. In vivo experiments showed that intraperitoneal administration of this engineered exosome-based peritoneal-localized hydrogel is useful against peritoneal dissemination of ovarian cancer [153]. The efficacy of repeated I-Pr chemotherapy is highly dependent on uniform distribution of the drug throughout the abdomen and deep penetration of the drug into the peritoneal tumor. In the future, development of drug modifications and improved delivery methods to enhance drug infiltration into peritoneal tumors may further improve the prognosis of patients with peritoneal dissemination.

## 6. Intratumor Route (IT)

Intratumorally administered drugs diffuse into the injected area and reach the targeted tumor in high concentrations. Next, the drug slowly moves from tissues into the systemic circulation over time and leads to early access to tumor-draining lymph nodes, which are important for anti-tumor immune responses [154]. In addition, IT may provide access to tertiary lymphoid structures that occur in the tumor microenvironment as a result of immune responses to tumor antigens [155,156]. Compared to conventional systemic chemotherapy, advantages of this method of administration not only include high intratumor and tumor tissue drug concentrations and early delivery to tumor-associated lymph nodes, but also reduced systemic toxicity [157]. Intratumoral administration began in the 19th century with local injections of *Streptococcus pyogenes* and Corey toxin for treatment of soft-tissue sarcomas [158], and has been used for intravesical injections of Bacillus Calmette–Guerin (BCG), which is still used today as a treatment for superficial bladder cancer [159,160]. Recent studies have clarified the importance of an anti-tumor immune response mediated by IFNγ-producing T cells and natural killer (NK) cells, as well as by macrophages activated as a result of BCG injection [161]. Therefore, intratumoral immunotherapy is currently being intensively studied. Intratumoral immune therapies inject immunostimulatory products directly into a tumor lesion to locally stimulate an antitumor immune response and to generate a systemic immune response against the tumor by immune cells and antibodies in the blood and lymph [162]. At present, clinical trials of intratumoral administration of many immunostimulatory products and combination therapy with other drugs are being conducted for various carcinomas, and favorable results have been reported, mainly for malignant melanoma (Table 6). However, while the dose of systemic chemotherapy is calculated based on a patient’s body weight and body surface area, a method for calculating doses for intratumoral administration has not yet been determined. Since it is necessary to consider the extent of the lesion, the size of the tumor, and effects of concomitant therapy, methods for calculating doses for intratumoral administration and determining the appropriate regimen are future issues [163] (Table 6).

Supplemental Appendix A shows the drugs described so far and their mechanisms of action [173,174,175,176,177,178,179,180,181,182,183,184,185,186,187,188,189,190,191,192,193].

## 7. Emerging Preclinical Strategies for Intrathoracic Administration Route via Lymphatics

Efficiency of anticancer drug delivery by IV administration to lymph nodes is poor compared to blood-rich organs such as liver and lung. Recently, we examined retrograde administration of PTX from lymphatic vessels as a novel route of administration for extensive lymph node metastases (ELM) in the abdomen. The thoracic duct, the body’s central lymphatic vessel, originates in the cisterna chyli in the retroperitoneum, ascends between the esophagus and the descending aorta in the mediastinum, and flows into the left venous angle in humans [194,195]. Therefore, retrograde administration via the thoracic duct may deliver high doses of anticancer agents to metastatic lymph nodes in the retroperitoneum with low systemic toxicity. Based on this hypothesis, we used a swine model to catheterize the thoracic ducts of pig necks and to infuse PTX via catheters. Then, we compared pharmacokinetics of PTX administered intrathoracically to those of intravenous infusion [196]. The concentration of PTX in serum, liver, and spleen was significantly lower following thoracic duct (IT) infusion than after intravenous (IV) administration, 1–8 h after drug infusion. However, PTX levels in abdominal lymph nodes were maintained at relatively high levels up to 24 h after IT infusion compared to after IV infusion. Therefore, IT delivery of PTX into the thoracic duct may yield clinical benefits for patients with ELM in abdominal malignancies.

## 8. Conclusions and Future Directions

This review presents a comprehensive overview of current perspectives on routes of drug administration for cancer. Recently, various types of drugs, such as antibody preparations and nanomicellar modifications, have been developed to enhance selective targeting to solid tumors. However, the route of drug administration is another critical determinant that can regulate pharmacokinetics and toxicity, impacting the clinical efficacy of anti-cancer drugs [197]. A recent report shows that anti-PD-L1 antibody can be delivered more efficiently by an intraperitoneal route compared with the EPR effect of systemic infusion [198]. Future studies need to examine pharmacodynamics of newly developed anti-cancer agents after administration by different routes.

## Figures and Tables

**Figure 1 pharmaceutics-15-01664-f001:**
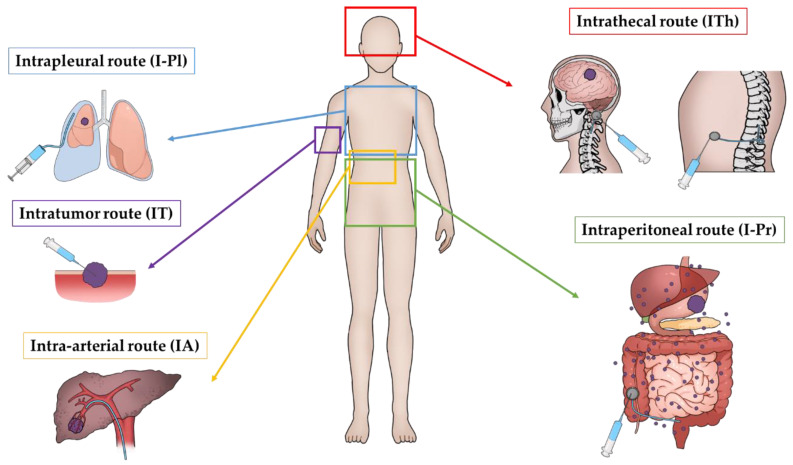
Schematic diagram of special anticancer drug administration route.

**Table 1 pharmaceutics-15-01664-t001:** Studies that administered chemotherapeutics by an intra-arterial route.

Indications	Type of Cancer	Anti-Cancer Drug	Study Phase
Liver tumor	Hepatocellular carcinoma	Cisplatin (TACE [18], HAIC [30,31,32,33,34,35])	-
		Doxorubicin (TACE [18], TACE-DEB [18], B-TACE [25])	-
		Miriplatin (TACE [18], B-TACE [25,26,28], HAIC [32])	-
		Epirubicin (TACE [18], B-TACE [25], HAIC [31,32])	-
		5-FU (HAIC [30,31,32,33,34,35])	-
		Mitomycin (HAIC [30,31,32])	-
		Oxaliplatin (HAIC [31])	-
Metastatic liver tumor	Colorectal cancer	Floxuridine [40]	Ⅱ NCT00492999
		Oxaliplatin [41]	Ⅱ/Ⅲ NCT02494973
Brain tumor	Non-GBM gliomas	Carboplatin [43]	-
	Primary central nervous system lymphoma (PCNSL) Primitive neuroectodermal tumor (PNET) Germ cell tumor	Methotrexate [44]	Ⅱ NCT00596154
		Carboplatin [44]	-
Head and neck cancer	Head and neck cancer	Cisplatin [46]	-
		Docetaxel [47]	-
		Nedaplatin [47]	-

**Table 2 pharmaceutics-15-01664-t002:** Studies that employed intrathecal drug administration and CED.

Indications	Type of Cancer	Anti-Cancer Drug	Study Phase
Brain tumor	Central nervous system (CNS) leukemia	Methotrexate [61]	-
		6-mercaptopurine [60]	-
		Vincristine [61]	-
		Cyclophosphamide [61]	-
	Metastatic medulloblastoma Ependymoma	CAR T cells [62]	-
Leptomeningeal metastasis	Breast cancer	Liposomal cytarabine [63]	Ⅲ NCT01645839
Brain tumor (CED)	Recurrent malignant glioma	Paclitaxel [70]	-
		Topotecan [70]	Ⅰ NCT03154996
		Tf-CRM107 [70]	Ⅲ NCT00076986
		TP-38 [70]	Ⅰ/Ⅱ NCT00074334
		IL13-PE38QQR [70]	Ⅲ NCT00076986
		Reovirus [70]	Ⅰ NCT02444546
	Recurrence glioblastoma (rGBM)	Paclitaxel [70]	-
		IL13-PE38QQR [70]	Ⅲ NCT00076986
		LIPO-HSV-1-tk [70]	-
		CpG-28 [70]	Ⅱ NCT05506969
	Recurrent malignant brain tumors	Tf-CRM107 [70]	-
		TP-38 [70]	Ⅰ/Ⅱ NCT00074334
	Recurrent high-grade glioma (HGG)	Topotecan [70]	Ⅰ NCT03154996
		Liposomal irinotecan [70]	Ⅰ NCT02022644
		AP-12009 [70]	Ⅲ NCT00761280
	Newly diagnosed malignant glioma	IL13-PE38QQR [70]	Ⅲ NCT00076986
	Malignant glioma	131I-chTNT-1/B MAb (Cotara) [70]	Ⅰ/Ⅱ NCT00509301

**Table 3 pharmaceutics-15-01664-t003:** Studies that employed intrapleural drug administration.

Indications	Type of Cancer	Anti-Cancer Drug	Study Phase
Malignant pleural effusion (MPE)	Nonsquamous non-small cell lung cancer (NS-NSCLC)	Bevacizumab [84]	-
	NSCLC	Bevacizumab [85]	-
	Lung cancer	Tumor microparticles packaging methotrexate (TMPs-MTX) [89]	-
	NSCLC	TMPs-MTX [90]	-

**Table 4 pharmaceutics-15-01664-t004:** Studies that employed intraperitoneal drug administration (heated intraperitoneal chemotherapy (HIPEC)).

Indications	Type of Cancer	Anti-Cancer Drug	Study Phase
Peritoneal metastasis (PM)	Colorectal cancer	Oxaliplatin [104]	-
	Colon cancer	Mitomycin-C [105]	Ⅳ NCT05250648
	Ovarian cancer	Carboplatin [106]	Ⅲ NCT00426257
	Ovarian cancer	Paclitaxel [106]	Ⅲ NCT00426257
	Gastric cancer	Docetaxel [110]	Ⅲ NCT03023436
	Gastric cancer	Mitomycin C [110]	Ⅲ NCT02158988
	Gastric cancer	Cisplatin [110]	Ⅲ NCT02158988
	Gastric cancer	Oxaliplatin [110]	Ⅲ NCT03348150
	Pancreatic cancer	Gemcitabine [111]	-
Primary and recurrent cancer	Ovarian cancer	Cisplatin [107]	Ⅲ NCT00426257
Primary cancer	Malignant peritoneal mesothelioma	Doxorubicin [108]	-
	Malignant peritoneal mesothelioma	Cisplatin [108]	-

**Table 6 pharmaceutics-15-01664-t006:** Studies that employed intratumor drug administration (IT).

Indications	Type of Cancer	Anti-Cancer Drug	Study Phase
Solid tumor	Melanoma	PV-10 [164]	Ⅱ NCT00521053
		SD-101 [165]	Ⅰ/Ⅱ NCT02521870
		Tilsotolimod [166]	Ⅰ/Ⅱ NCT02644967
		Talimogene laherparepvec (T-VEC) [167]	Ⅲ NCT00769704
		CAVATAK [168]	Ⅱ NCT01227551
		HF10 [169]	Ⅱ NCT02272855
	Primary Hepatocellular Carcinoma	Pexa-Vec [170]	Ⅱ NCT01171651
	Glioma grade IV	PVSRIPO [171]	Ⅰ NCT01491893
	Malignant glioma	DNX-2401 [172]	Ⅰ NCT00805376

## Data Availability

Not applicable.

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
