# Peer review of "Anatomical Targeting of Anticancer Drugs to Solid Tumors Using Specific Administration Routes: Review"

_pharmaceutics, 2023, doi:10.3390/pharmaceutics15061664_

Round 1

Reviewer 1 Report

The review article is well presented, and it is always interesting to read the article. It makes a very interesting read and really enjoyable approach. 

However the article missed schematic diagrams to summarize the review outlines.

The authors should add anew section about regional intratumoral injection of chemotherapeutic drugs in the form of drug delivery system such as implants.

moderate editing of English is required 

Reviewer 2 Report

In my opinion the paper is informative but too short.

A paragraph on convection enhanced delivery should be added for IT.

Authors should explain the choice of drugs used for local delivery.

In the Introduction Authors should cite the work of WC Chan's group on limited tumor nanoparticle delivery (eg https://doi-org.insb.bib.cnrs.fr/10.1021/acsnano.9b08142 ).

A schme of the different routes shoulmd be added as well as techniques (eg PIPAC).

Authors could add a section on emerging preclinical strategies.

Authors should describe the mechanisms of activity of the drugs cited.

Phases of trials and identifiers need to be added in Tables.

Authors need to report the response rates in the text or Tables.

Authors should explain CRS.

Reviewer 3 Report

This review summarizes clinical results and problems and discusses future directions of regional cancer therapy with local admin-istration of chemotherapeutants. This is a well-organized review article which is interesting, and can be published after minor modification.

1. For intraperitoneal (I-Pr) routes, the following recent literature might be useful, https://doi.org/10.1021/acsnano.3c00804

2. Heianna et al. suggested that selective intra-arterial chemoradi-otherapy with docetaxel and nedaplatin may achieve both good local control and survival in bulky node-fixed head and neck cancer of unknown primary origin (HNCUP). If possible, please explain why  selective intra-arterial chemoradi-otherapy with docetaxel and nedaplatin can result in such effect.

3. pre-clinical studies showed that PIPAC results in good distribution and penetration into tu-mor nodules in the abdominal cavity. Please explain why PIPAC enables good distribution and penetration.

4. The format of references should be uniform, for example the capitals in the title of the referred literatures are not uniform. 

Reviewer 4 Report

In this review Saito et al. summarized the methods of regional cancer therapy with local administration of chemotherapeutants. This review is written very well, well organized and comprehensively described. The discussed topic could significantly contribute to the field. I have only some minor comments:

1) Authors focused only on the methods and advantages of regional cancer therapy with local administration of chemotherapeutants but ignore the disadvantages of each method.

2) Authors should also compare between each method from the point of the most efficient, risk, cost, interference (invasiveness), and survival rate. These data will be so useful for readers.

3) I also suggest a diagram showing examples of localized chemotherapy administration.

4) Tables should be properly cited in suitable places within the manuscript. See for example Tables 1 and 2 which were found very so late in the text.

5) Font size of Table 1 title is very large. Please use the same font size throughout the whole manuscript.

6) Delete "※Superscripts correspond to reference numbers." beneath each table and just add the cited number within [  ] as in-text citations.

7) "Conclusions and Future Directions" is very short and could be improved
